# Contributions Regarding the Study of Pulsatility and Resistivity Indices of Uterine Arteries in Term Pregnancies—A Prospective Study in Bucharest, Romania

**DOI:** 10.3390/diagnostics14222556

**Published:** 2024-11-14

**Authors:** Giorgia Zampieri, Alexandra Matei, George Alexandru Roșu, Andrei Marin, Mircea Octavian Poenaru, Cringu Antoniu Ionescu

**Affiliations:** 1Faculty of Medicine, University of Medicine and Pharmacy Carol Davila, 020021 Bucharest, Romania; 2Department of Obstetrics and Gynecology, Saint Pantelimon Emergency Clinical Hospital, 021659 Bucharest, Romania; 3Department of Plastic Surgery, Saint Ioan Emergency Clinical Hospital, 042122 Bucharest, Romania; 4Bucur Maternity, Saint Ioan Emergency Clinical Hospital, 040292 Bucharest, Romania

**Keywords:** uterine artery Doppler, pulsatility index, resistivity index, high-risk pregnancy, intrauterine growth restriction, gestational hypertension, preeclampsia, gestational diabetes

## Abstract

Pregnancy is a complex stage in a woman’s life, considering the physical and psychological changes that occur. The introduction of Doppler studies of the pregnant woman’s vessels and those of the fetus has proven to be a useful tool in evaluating the maternal-fetal relationship. **Objective**: The study aims to assess the correlations of PI and RI values in term pregnancies. **Methods:** This analysis is based on the prospective evaluation of medical data from 60 patients who were admitted to the Obstetrics and Gynecology department of Saint Pantelimon Hospital in Bucharest, Romania, from May to August 2024. Among the examined parameters are patient age, blood pressure, amniotic fluid quantity, placenta location, and pulsatility and resistivity indices of uterine arteries. **Results:** A higher diastolic blood pressure is associated with higher mean PI and RI values, indicating that diastolic blood pressure has a significant correlation to these values. The mean RI shows a moderately negative and significant correlation, suggesting that a lower level of amniotic fluid is associated with a higher mean RI. Regarding the PI value of the uterine arteries, the *p*-value suggests that the difference between the groups with and without associated diseases is statistically significant. Placental insertion on the anterior or posterior uterine wall does not have a significant impact on the PI and RI values of the uterine arteries, but the values are higher in the contralateral part of the placental insertion. **Conclusions:** These results strengthen the evidence previously demonstrated. Uterine artery Doppler ultrasonography is an extremely useful tool in monitoring and managing high-risk pregnancies.

## 1. Introduction

An inadequate adjustment to the physical, psychological, and social changes during pregnancy can lead to the development of diseases, turning it into a high-risk obstetrical pregnancy with significant impact [1,2]. As there are no clear criteria for assessing the risk in pregnancy, which is the result of a subjective assessment by the patient or doctor [2,3], we can assert that the number of high-risk obstetrical pregnancies is increasing, directly proportional to the increase in the use of screening and diagnostic methods, and implicitly early treatment [3,4].

Normal blood flow in the feto-placental circulation is important for creating a healthy intrauterine environment through the proper functioning of the placenta, which ensures the harmonious growth of the fetus [5,6,7,8]. The relationship between abnormal Doppler velocimetry at the level of the uterine arteries and the appearance of various obstetric complications, such as preeclampsia and intrauterine growth restriction, has been intensively studied over the years, with important conclusions for obstetrical practice [7,9,10,11].

The Doppler study of the uterine arteries between weeks 11 and 14 of pregnancy has a predictive role for preeclampsia [12,13,14,15]. Although trophoblastic invasion cannot be directly evaluated, Doppler studies allow for non-invasive assessment of uteroplacental circulation by comparing systolic and diastolic waves [12,16,17]. The pulsatility index (PI), resistance index (RI), systolic/diastolic ratio (S/D), and the presence of a protodiastolic notch are indicators used in studying blood flow at the level of the uterine arteries [5,18,19,20]. One of the most commonly used indices is the pulsatility index (PI), which is defined as the difference between the maximum systolic flow velocity and the terminal diastolic velocity, divided by the average flow velocity over time [5,21,22,23]. The formula used for calculation is PI = (PSV − EDV)/TAV, where PSV represents the peak systolic velocity, EDV represents the end-diastolic velocity, and TAV represents the average flow velocity over time [21]. Low terminal diastolic velocities (EDVs) and the presence of a notch are frequently observed in non-pregnant women, or pregnant women in the first trimester [17,24,25]. Persistence of the notch or an abnormal ratio of velocities results from increased vascular impedance and is associated with inadequate trophoblastic invasion [26,27].

Some studies have reported that the continuously increasing resistance to flow through the uterine arteries observed in preeclampsia, intrauterine growth restriction, and small for gestational age fetuses (SGA = small for gestational age) is associated with unfavorable outcomes, such as intrauterine fetal death, cesarean section indicated by fetal distress, and low umbilical cord pH [18,25,28,29,30,31]. Recent studies show that in term pregnant women with late-onset preeclampsia, the pulsatility index of the uterine arteries began to increase towards the end of pregnancy, emphasizing the importance of assessing flow through the uterine arteries in the third trimester of pregnancy [5,24,27].

## 2. Materials and Methods

Recent studies have shown that modifications of PI and RI indices in term pregnancies may appear as a result of the development of an associated disease, which is why evaluating uterine artery Doppler in the third trimester has proven beneficial. The hypothesis of this study is based on the suspicion that uterine artery PI and RI indices correlate with certain aspects that can be evidenced in high-risk pregnancies. To confirm the working hypothesis, the following specific objectives were stated: (i) identification of patients’ anamnestic characteristics through detailed anamnesis and clinical examination, with evaluation of blood pressure and maternal weight; (ii) quantification of relevant variables—PI, RI of uterine arteries, objective evaluation of amniotic fluid quantity, position and maturity of placenta, as well as estimated fetal weight with the corresponding percentile and abdominal circumference percentile; (iii) identification of patients with associated pathologies and their descriptive analysis; (iv) evaluation of how uterine artery PI and RI correlate with various variables, such as age, weight, smoking status, systolic and diastolic blood pressure, gestational age, amniotic fluid quantity and position, and maturity degree of placenta; (v) evaluation of the correlation between uterine artery PI and RI and associated pathologies, namely identifying specific modifications depending on various conditions.

This study combines the outcomes of a prospective analysis of medical data from 60 pregnant patients with singleton live pregnancies, at term (with gestational ages >37 weeks and <41 weeks), admitted to the Obstetrics–Gynecology department of St. Pantelimon Emergency Clinical Hospital, Bucharest, Romania, during the period from 1 May to 31 August 2024.

The research plan was implemented in accordance with the norms of scientific, professional, and university ethics, in compliance with the provisions of the code of ethics and deontology of the involved institutions. The scientific research activity was previously approved by the Ethics Committee of the St. Pantelimon Emergency Clinical Hospital, Bucharest, Romania, as per decision no. 47/20 November 2023.

To create the necessary database, the inclusion criteria for patients in the study were:-age > 16 years;-gestational age between 37 and 41 weeks;-patients admitted to the Obstetrics–Gynecology department of the St. Pantelimon Emergency Clinical Hospital, Bucharest, Romania, during the period from 1 May to 31 August 2024;-adult patients who provided consent for the processing of medical data for scientific research purposes at the time of admission;-minor patients, aged >16 years, whose legal representative expressed consent for the processing of medical data for scientific research purposes at the time of admission.

Additionally, the exclusion criteria for the study were as follows:
-age < 16 years;-gestational age < 37 weeks;-multiple pregnancy;-patients with antepartum fetal death;-adult patients who refused the processing of medical data for scientific research purposes at the time of admission;-minor patients, aged >16 years, whose legal representative refused the processing of medical data for scientific research purposes at the time of admission.

The foundation of this research involved combining the data found in the general clinical observation sheet of the patients and their ultrasonographic findings obtained through transabdominal scans.

The elements of descriptive statistics highlighted the following components:Quantitative variables: age (completed years), weight, height, systolic and diastolic blood pressure values, gestational age, estimated ultrasonographic fetal weight, fetal weight percentile, and abdominal circumference percentile on ultrasound scan, PI and RI values of the left uterine artery, right uterine artery, and umbilical artery, average PI and RI values of the left and right uterine arteries, number of cigarettes/day, value of the single deepest pocket of amniotic fluid, Grannum maturity grade of the placenta;Dichotomous qualitative variables: presence of smoking—yes/no, position of the placenta—anterior/posterior uterine wall and right/left fundus;Nominal qualitative medical variables: assessment of the amount of amniotic fluid, presence of associated disease or conditions.

After collecting the necessary data, they were integrated into an Excel (version 16.90.2) database for statistical processing. For the statistical processing of the study data, IBM SPSS Statistics for Windows, Version 29.0 (30-day trial version) (IBM Corp., Armonk, NY, USA) was used. Nominal data were presented as absolute frequency and percentage, while continuous variables were expressed as mean, standard deviation, median, minimum, and maximum. The analysis of the association between categorical variables was performed using cross-tabulation and the χ^2^ (chi-square) test. If the results of the chi-square test were sufficiently altered to be considered unreliable, Fisher’s exact test was used. The degree of correlation (r) between the studied parameters was assessed by calculating the Pearson correlation coefficient. To compare means based on the dichotomous variables in the study, the *t*-test for independent samples was used. For comparing means of parameters between groups, ANOVA was used. A *p*-value of <0.05 was considered statistically significant.

## 3. Results

The first part presented in this section is represented by the descriptive statistical analysis. Thus, the first variable studied is age. Overall, this sample of 60 pregnant women has an average age of 23.63 years, with a relatively moderate variation in ages (ranging from 16 to 40 years).

The distribution of the weight of pregnant women varies significantly, with an average of approximately 76 kg, showing considerable weight variation, highlighted by the large difference between the minimum and maximum values (56 kg and 132 kg). The distribution of the height of pregnant women has an average of approximately 163.55 cm, and the values are relatively well distributed around this average, with a standard deviation of 6.657 cm. The extreme values (147 cm and 180 cm) indicate a significant diversity in height.

Systolic blood pressure values range between 106 and 147 mmHg, with an average value of 119.33 mmHg, which generally indicates a normal systolic blood pressure, but with some cases of hypertension highlighted. The graphical representation of these data can be seen in Figure 1.

The diastolic blood pressure values range between 57 and 98 mmHg, with an average of 69.62 mmHg. The curve of the distribution of diastolic blood pressure values is illustrated in Figure 2.

Approximately 54.2% of patients do not smoke, while 45.8% are smokers, with an average consumption of about 10 cigarettes per day, showing significant variations. Some women smoke as few as two cigarettes per day, while those with the highest consumption reach up to 30 cigarettes per day.

Most gestational ages are validated at 37 and 39 weeks, with a significant frequency for each. The minimum value is 37 weeks, while the maximum is 40 weeks. The median and mean are very close, suggesting a fairly uniform distribution around the average value of 38 weeks.

The mean pulsatility index (PI) for the right uterine artery is 0.8015. The median represents the middle value of the distribution, so half of the individuals have a PI below 0.7300 and half above this value. The slight difference between the mean and median suggests a possible mild asymmetry in the distribution, indicating that there are a few higher values that increase the mean. The standard deviation measures the variability of the PI values in the sample. With a value of 0.29188, there is a moderate dispersion around the mean, indicating that there are some variations in blood flow of the right uterine artery among individuals. The sample shows moderate variability in the PI of the right uterine artery, with values ranging from 0.44 to 1.95.

The mean pulsatility index (PI) for the left uterine artery is 0.8107. The median indicates that half of the individuals have a PI lower than 0.7550 and half have higher values. The slight difference between the mean and median indicates a relatively balanced distribution, but with a slight asymmetry towards higher values. The distribution of PI values for the left uterine artery is similar to that for the right artery, having a slightly higher mean (0.8107 compared to 0.8015). The maximum value of the mean PI (1.34) is lower than the maximum observed for individual uterine arteries, indicating that the mean tends to be influenced by intermediate values. The distribution is relatively balanced, with a significant percentage (45%) of individuals having a PI above the normal limit (according to international nomograms). The distribution of mean PI values is illustrated in Figure 3.

The resistivity index (RI) of the right uterine artery in this sample shows moderate variability, with values concentrated around the mean of 0.5068. The minimum and maximum values (ranging from 0.33 to 0.78) reflect a relatively narrow range of blood flow resistance in this artery, without major extremes. The resistivity index (RI) of the left uterine artery shows low variability, with values ranging from 0.29 to 0.77. The small differences between the means and medians suggest a symmetric distribution, and the standard deviation indicates relatively low dispersion of the RI values. The minimum and maximum values of the mean RI range from 0.36 to 0.69. The distribution of mean RI values can be observed in Figure 4.

The mean and median values of the pulsatility index (PI) and resistivity index (RI) of the umbilical artery suggest a balanced distribution.

Analyzing the amount of amniotic fluid, 81.7% of pregnant women have a normal amount, while 10% are diagnosed with polyhydramnios and 8.3% with oligohydramnios. These percentages are represented in Figure 5. The quantitative assessment of the amniotic fluid volume was performed by measuring the deepest vertical pocket (DVP), with these values showing considerable variability in the amount of amniotic fluid. The mean of 5.023 cm suggests a moderate amount, but with a high standard deviation (2.2440 cm). The difference between the mean and median indicates a slightly asymmetric distribution, with some higher values influencing the mean.

Based on the insertion of the placenta on the anterior or posterior uterine wall, there is a slight predominance of the anterior position (51.7%). The distribution of placental positions between the left and right uterine sides is balanced, with a slight predominance of the position in the right uterine side (51.7%). In total, 63.3% of pregnant women have a placenta with a maturity grade of Grannum 3, while the rest have a placenta with a maturity grade of Grannum 2.

In the studied sample, 51.7% of pregnant women have an associated disease or special conditions, indicating a diversity of issues that can influence pregnancy. The high percentage of polyhydramnios (16.15%) and intrauterine growth restriction (19.35%) suggests that these conditions are relatively frequent in this sample and require careful monitoring. Other conditions, such as gestational diabetes, hypertension, and small or large fetal weight, are also present, but with more varied prevalences, represented graphically in Figure 6.

### 3.1. Correlations Between Mean PI and RI of Uterine Arteries and Age

Regarding the statistical analysis between mean PI and age, the Pearson correlation of −0.068 suggests a very weak negative correlation between the patient’s age and mean PI value, with a *p*-value of 0.604, which is much greater than the typical significance value of 0.05.

As for the mean RI, the Pearson correlation of −0.043 suggests an extremely weak negative correlation between the patient’s age and mean RI. The *p*-value of 0.742 is much greater than the typical significance value of 0.05, indicating that the result is not statistically significant.

Thus, there is no significant correlation between the patients’ age and mean PI and RI values. Both the correlation coefficient values and the *p*-values suggest that variations in age are not associated with variations in mean PI or RI.

### 3.2. Correlations Between Mean PI and RI of Uterine Arteries and Weight

The Pearson correlation of −0.040 suggests a very weak negative correlation between the pregnant woman’s weight and mean PI, with a *p*-value of 0.760. The Pearson correlation of −0.002 suggests an extremely weak and almost null correlation between the pregnant woman’s weight and mean RI, with *p* = 0.989, much greater than the typical significance value. Therefore, there are no significant correlations between the pregnant woman’s weight and mean PI and RI values, judging by the small correlation coefficient values and the very large *p*-values.

### 3.3. Correlations Between Mean PI and RI of Uterine Arteries and Systolic Blood Pressure

The following data reflect the correlations between the systolic blood pressure and the two mean values of uterine blood flow—mean PI and mean RI.

Regarding the mean PI value, the Pearson correlation of 0.219 suggests a weak positive correlation between the systolic blood pressure and mean PI. This indicates that, in general, a higher systolic blood pressure value might be associated with a higher mean PI value, but the relationship is weak. The *p*-value of 0.093 is greater than the typical significance threshold of 0.05, meaning that this correlation is not statistically significant. Therefore, there is not enough evidence to claim that the systolic blood pressure influences mean PI significantly.

On the other hand, the Pearson correlation of 0.317 suggests a moderate positive correlation between the value of systolic blood pressure and mean RI. This coefficient indicates that, generally, a higher systolic blood pressure is associated with a higher mean RI. With a *p*-value of 0.014, this correlation is statistically significant.

Thus, there is sufficient evidence to support the claim that the systolic blood pressure influences mean RI in a significant manner.

### 3.4. Correlations Between Mean PI and RI of Uterine Arteries and Diastolic Blood Pressure

The data below reflect the correlations between the diastolic blood pressure and the values of mean PI and RI. Regarding the mean PI value, the Pearson correlation of 0.366 suggests a moderate positive correlation, with a *p*-value of 0.004, indicating that this correlation is statistically significant. Similarly, the Pearson correlation of 0.430 suggests a moderate positive correlation between the value of diastolic blood pressure and mean RI. The *p*-value of 0.001 is much smaller than the significance threshold of 0.05, suggesting that this correlation is also statistically significant.

Concluding, the diastolic blood pressure has significant and positive correlations with both mean PI and RI. This suggests that a higher diastolic blood pressure is associated with higher values of mean PI and RI, indicating that diastolic blood pressure has a significant impact on these values.

### 3.5. Correlations Between Mean PI and RI of Uterine Arteries and Smoking

The data analyzed suggest that smokers have higher mean PI and RI values compared to non-smokers. This indicates a trend that smoking might be associated with higher values of PI. However, the analysis suggests that, although there are differences in the mean PI and RI values between smokers and non-smokers, these differences are not statistically significant according to the *t*-tests performed. This demonstrates that smoking does not have a significant impact on these values.

### 3.6. Correlations Between Mean PI and RI of Uterine Arteries and Gestational Age

As gestational age increases, PI and RI values decrease, as demonstrated by the Pearson correlation of −0.058 and −0.073, respectively, suggesting a weak negative correlation. However, the differences observed within term pregnancies are not significant.

### 3.7. Correlations Between Mean PI and RI of Uterine Arteries and Amniotic Fluid Quantity

The following data compare mean PI and RI between groups classified according to the amount of amniotic fluid: normal, polyhydramnios, and oligohydramnios.

Regarding mean PI, the median value is highest in the group with decreased amniotic fluid (0.9880) compared to the other groups. The mean PI value in the group of patients with polyhydramnios is the lowest (0.7383), while in the case of those with a normal amount of amniotic fluid, it is 0.7958. As for mean RI, the highest values (0.6080) are observed in the oligohydramnios group, and the lowest values (0.4725) are in the polyhydramnios group. These data can be consulted in Table 1.

By subsequent statistical analysis using an ANOVA test, in relation to mean PI, a *p*-value of 0.134 is obtained, suggesting that the differences in mean PI between groups are not statistically significant. In contrast, analyzing the mean RI value, a *p*-value of 0.011 suggests that the differences in mean RI between groups are statistically significant. This indicates that amniotic fluid volume has a significant effect on mean RI.

In conclusion, mean PI does not show significant differences between groups depending on amniotic fluid volume, while mean RI shows significant differences between groups, suggesting that amniotic fluid volume significantly influences mean resistivity index.

The following data analyze the correlations between amniotic fluid volume, measured by the single deepest pocket (SDP) of amniotic fluid detected ultrasonographically, and the two mean PI and RI values. As SDP increases, mean PI tends to decrease slightly, but with a *p*-value of 0.184. Therefore, there is not enough evidence to support that amniotic fluid has a significant impact on mean PI. Regarding mean RI, the Pearson correlation of −0.258 suggests a moderately negative correlation between the fluid pocket value and mean RI. This indicates that, in general, as the fluid pocket value increases, mean RI tends to decrease. The *p*-value of 0.046 is less than the significance threshold of 0.05, suggesting that this correlation is statistically significant. Thus, there is sufficient evidence to assert that amniotic fluid has a significant impact on mean RI, with a moderately negative influence.

In conclusion, mean PI does not show a significant correlation with amniotic fluid volume, while mean RI shows a moderately negative and statistically significant correlation, suggesting that a lower level of amniotic fluid is associated with a higher resistivity index. These results reinforce the previous findings.

### 3.8. Correlations Between Mean PI and RI of the Uterine Arteries and the Grannum Maturity Grade of the Placenta

The mean PI is higher in the group with grade 3 Grannum maturity (0.8454) compared to grade 2 (0.7382), suggesting that mean PI tends to be higher in cases of more advanced placental maturity. The mean RI is slightly higher in the group with grade 3 Grannum maturity (0.5195) compared to grade 2 (0.4916), suggesting a tendency for mean RI to be higher in cases of more advanced placental maturity.

Analyzing mean PI, the *t*-test indicates a *p*-value of 0.041 for the assumption of equal variances, which is less than the significance threshold of 0.05. This suggests that there is a significant difference between mean PI for grade 2 and grade 3 placental maturity. Therefore, a more advanced placental maturity is associated with a higher mean PI. Correlating maturity grades with mean RI, the *t*-test indicates a *p*-value of 0.209 (for equal variances) and 0.175 (for the assumption of equal variances), both of which are greater than the significance threshold of 0.05. In conclusion, mean PI differs significantly between grade 2 and grade 3 Grannum placental maturity, with a higher mean pulsatility index observed in advanced placental maturity.

### 3.9. Correlations Between Mean PI of Uterine Arteries and PI of Umbilical Artery

The correlation of 0.255 suggests a moderately positive correlation between mean PI of uterine arteries and PI of umbilical artery. The *p*-value of 0.049 is less than the significance threshold of 0.05. This suggests that, within this dataset, a higher PI of the umbilical artery is associated with a higher mean PI, and this relationship is statistically significant.

### 3.10. Correlations Between PI and RI of Uterine Arteries and Placental Position

For all PI and RI values of right and left uterine arteries, no significant differences were found between patients with anterior placental insertion and those with posterior placental insertion, given that the *p*-value is constantly greater than the significance threshold of 0.05. This suggests that anterior or posterior placental insertion does not have a significant impact on these values.

Regarding placental localization on the right or left uterine side, for all PI and RI measurements of the right and left uterine arteries, significant differences were found between groups with positioning on the left and right uterine side, as follows:-Right uterine artery PI: The mean PI is significantly lower for the right uterine side location compared to the left;-Right uterine artery RI: The mean RI is significantly lower for the right uterine side location compared to the left;-Left uterine artery PI: The mean PI is significantly lower for the left uterine side location compared to the right;-Left uterine artery RI: The mean RI is significantly lower for the left uterine side location compared to the right.

More specifically, the mean PI and RI values of the right uterine artery are significantly lower for the right uterine side insertion of placenta compared to the left, while the mean PI and RI values of the left uterine artery are significantly lower for the left uterine side location of placenta compared to the right, as illustrated in Table 2. Concluding, these data demonstrate that PI and RI values are higher in the contralateral side of the placental insertion.

### 3.11. Correlations Between PI and RI of Uterine Arteries and Associated Conditions

The data below analyze the differences between groups (with and without associated diseases) for various measures of uterine blood flow, including PI and RI for the right and left uterine arteries, as well as their mean values, results which are demonstrated in Table 3.

Regarding the PI value of the right uterine artery, the *p* = 0.024 value suggests that the difference between the groups with and without diseases is statistically significant. PI value is higher in the group of patients with associated diseases. Referring to the PI value of the left uterine artery, a *p* = 0.312 value was found. However, when analyzing the mean PI value, we observe a *p*-value of 0.033, which indicates a significant difference between the groups, with a higher mean PI value in the group with associated diseases. Analyzing the RI value of the right uterine artery, a *p* = 0.051 value is found, suggesting a marginal difference between the groups. In the case of the RI and mean RI values of the left uterine artery, *p*-values above the statistical significance threshold of 0.05 were found. These results suggest that associated conditions may significantly influence the values of the indices.

The following data provide a detailed picture of PI and RI values for right and left uterine arteries, depending on different associated conditions included in the study.

-PI right uterine artery: F: 1.004 (*p* = 0.478). Differences between groups are not statistically significant;-PI left uterine artery: F: 1.073 (*p* = 0.429). Differences between groups are not statistically significant;-Mean PI: F: 0.882 (*p* = 0.572). Differences between groups are not statistically significant;-RI right uterine artery: F: 1.255 (*p* = 0.320). Differences between groups are not statistically significant;-RI left uterine artery: F: 0.842 (*p* = 0.604). Differences between groups are not statistically significant;-Mean RI: F: 1.088 (*p* = 0.419). Differences between groups are not statistically significant.

In conclusion, PI and RI values vary significantly between different associated diseases, as can be observed in the data analysis of Table 4. There are cases with extremely variable values, especially in conditions with a small number of cases. Regarding the ANOVA test results detailed in Table 5, no significant differences are detected between groups for any of the PI and RI measurements. This suggests that, from the perspective of these statistics, there is not enough evidence to support the idea that differences in PI and RI are preferentially associated with the types of pathologies studied. These results may also indicate that the observed variations may be influenced by other factors not analyzed in this context.

In conclusion, mean PI and RI values are strongly influenced by multiple factors, such as systolic blood pressure, diastolic blood pressure, amniotic fluid quantity, placental maturity, and the presence of associated pathologies. Elevated values have been observed in patients with hypertension, oligohydramnios, Grannum 3 placenta grading, and those experiencing associated diseases in pregnancy. Additionally, an interesting result is that the values of the indices are increased on the contralateral side of the placental insertion. All these findings are summarized in Table 6.

## 4. Discussion

There is a significant statistical correlation between both the systolic blood pressure and the mean RI value of the uterine arteries, as well as between the diastolic blood pressure and the mean PI and RI values. There is evidence supporting that trophoblast invasion is maximal in the first trimester, which is why Doppler evaluation of the uterine arteries is recommended as early as the first trimester of pregnancy—between 11 weeks of gestation and 13 weeks and 6 days of gestation [13,32]. Continuous monitoring of patients through Doppler studies and measurement of indices throughout pregnancy has proven useful, as it has been demonstrated that patients with persistently elevated PI have a higher risk of developing preeclampsia [33,34,35]. This is due to an association between the continuous deterioration of blood flow through the uterine arteries and gestational hypertension, independent of the PI measured in the second trimester of pregnancy [5,16,20].

Secondly, there is a moderate and significant negative correlation between the amount of amniotic fluid and the average RI value. The association between oligohydramnios and high RI reflects the complex interplay between amniotic fluid levels, placental health, and fetal well-being [36,37]. Oligohydramnios can indicate compromised placental function, which may lead to inadequate blood supply to the fetus. This can cause the blood vessels to constrict, increasing the RI as the body attempts to maintain adequate perfusion under stress [5,36]. A part of the pathophysiological mechanism is also explained by the fact that average PI value differs significantly between grade 2 and grade 3 of Grannum placental maturity, with higher values observed in the more advanced grade [38,39,40].

Regarding the placenta location, the insertion of the placenta on the anterior or posterior uterine wall does not have a significant impact on the PI and RI values of the uterine arteries, while the PI and RI values of the uterine arteries are significantly higher on the contralateral side of the placenta location, findings also demonstrated by Song et al. and Vaillant et al. [41,42]. The uterine artery on the contralateral side may have higher vascular resistance due to factors such as less demand for blood flow or compensatory mechanisms in response to placental positioning [42]. This increased resistance contributes to a higher PI [17,22,42].

Associated conditions can significantly influence the PI and RI values of the uterine arteries, with higher PI and RI values demonstrated in the group with associated diseases. In this study, there is insufficient evidence to support that the differences in PI and RI values are preferentially associated with the types of conditions studied; these results may indicate that there are many unexamined factors that can influence the observed variations. Conditions such as preeclampsia, gestational hypertension, or placental insufficiency can lead to increased vascular resistance in the uterine arteries [9,35,43]. Also, when the fetus is compromised due to placental dysfunction or other pregnancy-related issues, the body may respond by increasing resistance in the uterine arteries to prioritize blood flow to vital organs [5,38]. Higher PI values in the uterine arteries are associated with increased PI values in the umbilical artery [44].

There is no statistically significant correlation between age and the PI and RI values of the uterine arteries. The weight of the pregnant woman alone does not have a significant impact on the PI and RI values of the uterine arteries, but Cody et al. demonstrated that maternal BMI may be considered an additional risk factor [45].

Although the data suggest a trend that smoking may be associated with higher average PI values, a fact demonstrated also by Baki et al. [46], the differences are not significant. The present study’s limitations revolve around its exclusive focus on a single institution and the relatively small sample size. The sample size and the effect of reduced size limit the power of this study—a larger sample would be necessary to achieve a power of 0.80, sufficient to detect the effect. For detecting such subtle differences, a future study should be larger in order to achieve the desired power.

## 5. Conclusions

In conclusion, Doppler velocimetry study of the uterine arteries can be used to predict obstetric complications such as preeclampsia, intrauterine growth restriction (IUGR), and fetal death in utero. A substantial association exists between uterine artery PI and RI and various factors such as blood pressure, amniotic fluid volume, placental maturity, and concomitant conditions during pregnancy. Notably, elevated values have been observed in patients with hypertension, oligohydramnios, Grannum 3 placenta maturity, and those experiencing associated diseases in pregnancy.

Monitoring the PI of the uterine arteries can provide important insights into the health of the placenta and the fetus, helping to identify potential complications in pregnancies associated with various conditions. By incorporating new insights, a more meticulous and comprehensive strategy can be adopted in the management of these patients, reducing the maternal-fetal morbidity and mortality.

The results of the study can add to the existing knowledge base, supporting future research and improving the understanding of the physiological mechanisms involved in pregnancy.

## Figures and Tables

**Figure 1 diagnostics-14-02556-f001:**
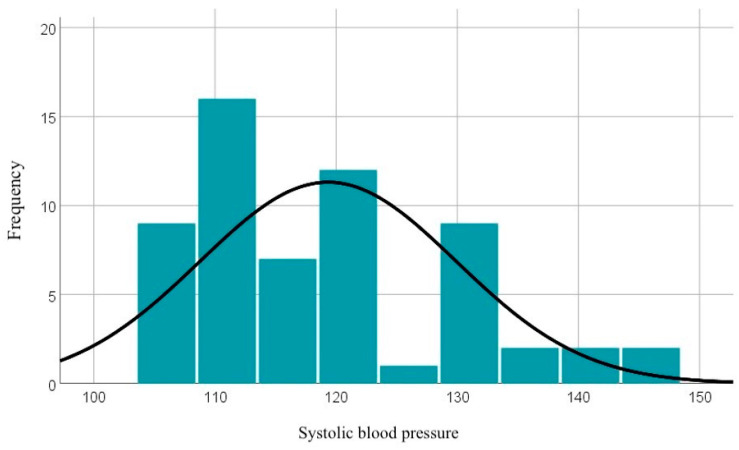
Graphical representation of systolic blood pressure values.

**Figure 2 diagnostics-14-02556-f002:**
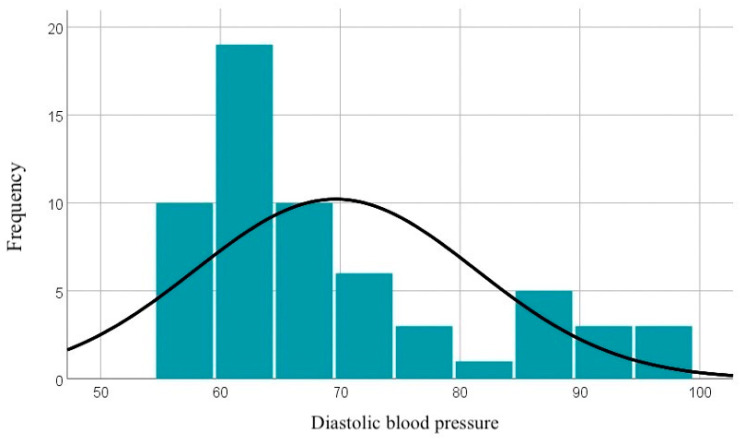
Graphical representation of diastolic blood pressure values.

**Figure 3 diagnostics-14-02556-f003:**
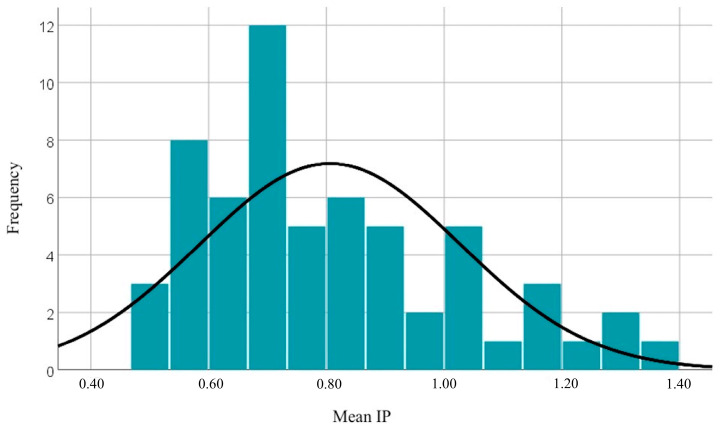
Graphical representation of mean PI values.

**Figure 4 diagnostics-14-02556-f004:**
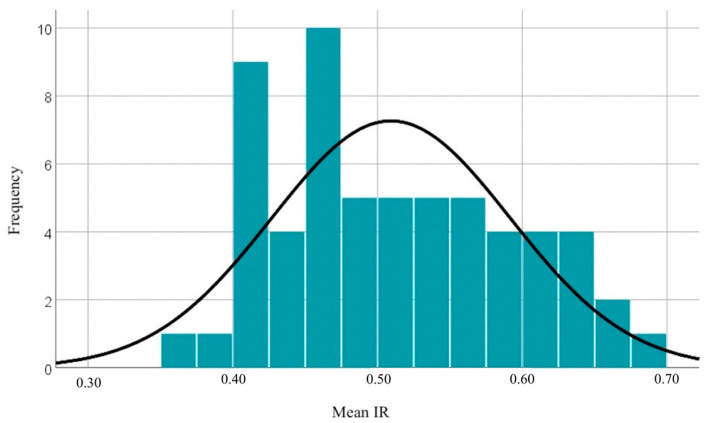
Distribution of mean RI values.

**Figure 5 diagnostics-14-02556-f005:**
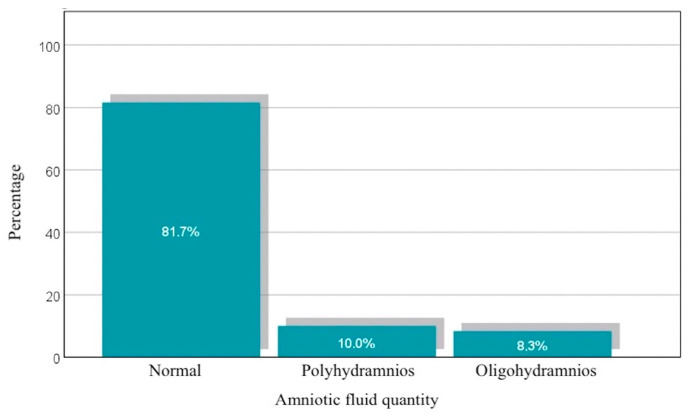
Distribution of patients based on the amount of amniotic fluid.

**Figure 6 diagnostics-14-02556-f006:**
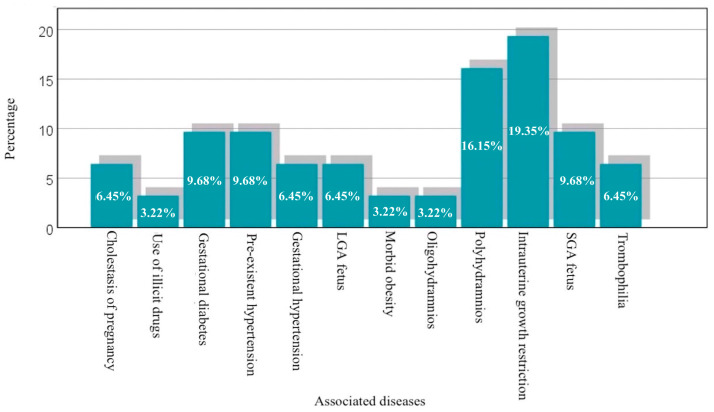
Graphical representation of cases with associated diseases.

**Table 1 diagnostics-14-02556-t001:** Descriptive statistics of mean PI and RI values for the three groups classified by amniotic fluid volume: normal, polyhydramnios, and oligohydramnios.

		N	Mean	Standard Deviation	Standard Error
Mean PI	Normal	49	0.7958	0.22720	0.03246
	Polyhydramnios	6	0.7383	0.15263	0.06231
	Oligohydramnios	5	0.9880	0.17315	0.07744
	Total	60	0.8061	0.22217	0.02868
Mean RI	Normal	49	0.5037	0.08245	0.01178
	Polyhydramnios	6	0.4725	0.04287	0.01750
	Oligohydramnios	5	0.6080	0.04040	0.01807
	Total	60	0.5093	0.08238	0.01064

**Table 2 diagnostics-14-02556-t002:** Correlations between PI and RI of uterine arteries and placental position—*t*-test.

		Levene’s Test for Equality of Variances		*t*-Test for Equality of Means		
		F	*p*	*t*	df	*p*
Right uterine artery PI	Assumption of Equal Variances	3.771	0.057	3.330	58	0.002
	Non-Assumption of Equal Variances			3.286	48.205	0.002
Right uterine artery RI	Assumption of Equal Variances	0.307	0.582	2.687	58	0.009
	Non-Assumption of Equal Variances			2.685	57.569	0.009
Left uterine artery PI	Assumption of Equal Variances	1.463	0.231	−2.248	58	0.028
	Non-Assumption of Equal Variances			−2.266	56.429	0.027
Left uterine artery RI	Assumption of Equal Variances	0.141	0.709	−2.134	58	0.037
	Non-Assumption of Equal Variances			−2.140	57.996	0.037

**Table 3 diagnostics-14-02556-t003:** Correlations between PI and RI of uterine arteries and associated conditions—*t*-test.

		Levene’s Test for Equality of Variances		*t*-Test for Equality of Means		
		F	*p*	t	df	*p*
Right uterine artery PI	Assumption of Equal Variances	3.128	0.082	−2.315	58	0.024
	Non-Assumption of Equal Variances			−2.341	53.870	0.023
Left uterine artery PI	Assumption of Equal Variances	4.158	0.046	−1.019	58	0.312
	Non-Assumption of Equal Variances			−1.030	54.708	0.308
Mean PI	Assumption of Equal Variances	2.694	0.106	−2.185	58	0.033
	Non-Assumption of Equal Variances			−2.202	56.472	0.032
Right uterine artery RI	Assumption of Equal Variances	1.509	0.224	−1.995	58	0.051
	Non-Assumption of Equal Variances			−2.005	57.628	0.050
Left uterine artery RI	Assumption of Equal Variances	1.137	0.291	−0.937	58	0.353
	Non-Assumption of Equal Variances			−0.942	57.356	0.350
Mean RI	Assumption of Equal Variances	0.068	0.795	−1.867	58	0.067
	Non-Assumption of Equal Variances			−1.869	57.913	0.067

**Table 4 diagnostics-14-02556-t004:** Correlations between PI and RI of uterine arteries and specific conditions—descriptive analysis.

		N	Mean	Standard Deviation	Standard Error
Right uterine artery PI	Cholestasis of pregnancy	2	0.5850	0.20506	0.14500
	Use of illicit drugs	1	1.2600		
	Gestational diabetes	3	0.8633	0.20841	0.12032
	Pre-existing hypertension	3	0.7267	0.20502	0.11837
	Gestational hypertension	2	1.1600	0.01414	0.01000
	Large for gestational age fetus	2	1.0300	0.52326	0.37000
	Class III obesity	1	0.7600		
	Oligohydramnios	1	1.0900		
	Polyhydramnios	5	0.6560	0.08764	0.03919
	Intrauterine growth restriction	6	0.8550	0.24485	0.09996
	Small for gestational age fetus	3	1.1067	0.74460	0.42990
	Thrombophilia	2	1.1050	0.26163	0.18500
	Total	31	0.8829	0.32428	0.05824
Left uterine artery PI	Cholestasis of pregnancy	2	0.6250	0.14849	0.10500
	Use of illicit drugs	1	1.2100		
	Gestational diabetes	3	1.2733	0.67337	0.38877
	Pre-existing hypertension	3	0.7733	0.36116	0.20851
	Gestational hypertension	2	0.7650	0.20506	0.14500
	Large for gestational age fetus	2	0.6600	0.02828	0.02000
	Class III obesity	1	0.5900		
	Oligohydramnios	1	0.9300		
	Polyhydramnios	5	0.8800	0.24839	0.11109
	Intrauterine growth restriction	6	0.9283	0.22868	0.09336
	Small for gestational age fetus	3	0.5933	0.04509	0.02603
	Thrombophilia	2	0.7650	0.43134	0.30500
	Total	31	0.8468	0.32305	0.05802
Mean PI	Cholestasis of pregnancy	2	0.6050	0.17678	0.12500
	Use of illicit drugs	1	1.2350		
	Gestational diabetes	3	1.0683	0.41552	0.23990
	Pre-existing hypertension	3	0.7500	0.21254	0.12271
	Gestational hypertension	2	0.9625	0.10960	0.07750
	Large for gestational age fetus	2	0.8450	0.24749	0.17500
	Class III obesity	1	0.6750		
	Oligohydramnios	1	1.0100		
	Polyhydramnios	5	0.7680	0.15007	0.06711
	Intrauterine growth restriction	6	0.8917	0.11303	0.04615
	Small for gestational age fetus	3	0.8500	0.39411	0.22754
	Thrombophilia	2	0.9350	0.34648	0.24500
	Total	31	0.8648	0.23800	0.04275
Right uterine artery RI	Cholestasis of pregnancy	2	0.4200	0.11314	0.08000
	Use of illicit drugs	1	0.6900		
	Gestational diabetes	3	0.5333	0.07767	0.04485
	Pre-existing hypertension	3	0.4900	0.09539	0.05508
	Gestational hypertension	2	0.6050	0.02121	0.01500
	Large for gestational age fetus	2	0.5950	0.17678	0.12500
	Class III obesity	1	0.5100		
	Oligohydramnios	1	0.6200		
	Polyhydramnios	5	0.4460	0.03362	0.01503
	Intrauterine growth restriction	6	0.5250	0.09138	0.03731
	Small for gestational age fetus	3	0.5667	0.19425	0.11215
	Thrombophilia	2	0.6400	0.07071	0.05000
	Total	31	0.5319	0.10769	0.01934
Left uterine artery RI	Cholestasis of pregnancy	2	0.4800	0.01414	0.01000
	Use of illicit drugs	1	0.6500		
	Gestational diabetes	3	0.6367	0.21385	0.12347
	Pre-existing hypertension	3	0.5233	0.19630	0.11333
	Gestational hypertension	2	0.5400	0.02828	0.02000
	Large for gestational age fetus	2	0.4400	0.00000	0.00000
	Class III obesity	1	0.4100		
	Oligohydramnios	1	0.5900		
	Polyhydramnios	5	0.5160	0.07197	0.03219
	Intrauterine growth restriction	6	0.5600	0.09466	0.03864
	Small for gestational age fetus	3	0.4200	0.02000	0.01155
	Thrombophilia	2	0.5000	0.18385	0.13000
	Total	31	0.5242	0.11564	0.02077
Mean RI	Cholestasis of pregnancy	2	0.4500	0.06364	0.04500
	Use of illicit drugs	1	0.6700		
	Gestational diabetes	3	0.5850	0.13611	0.07858
	Pre-existing hypertension	3	0.5067	0.10516	0.06071
	Gestational hypertension	2	0.5725	0.02475	0.01750
	Large for gestational age fetus	2	0.5175	0.08839	0.06250
	Class III obesity	1	0.4600		
	Oligohydramnios	1	0.6050		
	Polyhydramnios	5	0.4810	0.04189	0.01873
	Intrauterine growth restriction	6	0.5425	0.03313	0.01352
	Small for gestational age fetus	3	0.4933	0.10693	0.06173
	Thrombophilia	2	0.5700	0.12728	0.09000
	Total	31	0.5281	0.08182	0.01470

**Table 5 diagnostics-14-02556-t005:** Correlations between PI and RI of uterine arteries and specific conditions—ANOVA test.

		Sum of Squares	df	Quadratic Mean	F	*p*
Right uterine artery PI	Between groups	1.160	11	0.105	1.004	0.478
	Within groups	1.995	19	0.105		
	Total	3.155	30			
Left uterine artery PI	Between groups	1.200	11	0.109	1.073	0.429
	Within groups	1.931	19	0.102		
	Total	3.131	30			
Mean PI	Between groups	0.575	11	0.052	0.882	0.572
	Within groups	1.125	19	0.059		
	Total	1.699	30			
Right uterine artery RI	Between groups	0.146	11	0.013	1.255	0.320
	Within groups	0.202	19	0.011		
	Total	0.348	30			
Left uterine artery RI	Between groups	0.132	11	0.012	0.842	0.604
	Within groups	0.270	19	0.014		
	Total	0.401	30			
Mean RI	Between groups	0.078	11	0.007	1.088	0.419
	Within groups	0.123	19	0.006		
	Total	0.201	30			

**Table 6 diagnostics-14-02556-t006:** Summary table.

Mean PI and RI of uterine arteries and patient age	No significant correlation.
Mean PI and RI of uterine arteries and weight	No significant correlation.
Mean PI and RI of uterine arteries and systolic blood pressure	Systolic blood pressure influences mean RI in a significant manner—a higher SBP is associated with a higher mean RI.
Mean PI and RI of uterine arteries and diastolic blood pressure	Diastolic blood pressure has significant and positive correlations with both mean PI and RI—a higher DBP is associated with higher mean PI and RI.
Mean PI and RI of uterine arteries and smoking	There is a trend that smoking might be associated with higher PI values, but the difference is not statistically significant.
Mean PI and RI of uterine arteries and gestational age	As gestational age increases, PI and RI of uterine arteries decrease.
Mean PI and RI of uterine arteries and amniotic fluid quantity	Low level of amniotic fluid is associated with higher RI in uterine arteries.
Mean PI and RI of uterine arteries and placental maturity	Higher mean PI is observed in patients with advanced placental maturity—grade 3 Grannum.
Mean PI of uterine arteries and PI of umbilical artery	A higher PI of the umbilical artery is associated with a higher mean PI of uterine arteries.
Mean PI and RI of uterine arteries and placental position—anterior/posterior wall	Anterior or posterior placental insertion does not have a significant impact on PI and RI values.
Mean PI and RI of uterine arteries and placental position—left/right	PI and RI values are higher in the contralateral side of the placental insertion.
Mean PI and RI of uterine arteries and associated conditions	PI value is higher in the group of patients with associated pathology. PI and RI values vary significantly between different associated diseases.

## Data Availability

The raw data supporting the conclusions of this article will be made available by the authors on request.

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
