# Peer review of "Contributions Regarding the Study of Pulsatility and Resistivity Indices of Uterine Arteries in Term Pregnancies—A Prospective Study in Bucharest, Romania"

_diagnostics, 2024, doi:10.3390/diagnostics14222556_

Round 1

Reviewer 1 Report

Comments and Suggestions for Authors

Doppler studies in obstetrics have been around for a couple of decades.In this paper PI and RI doppler studies are done in a very detail mode as considering plasental location,amniotic volume,blood pressure. Actually this hard work does not add too much on current understanding of doppler studies. On the other hand ,there is scentific confirmation of some pregnancy problems and doppler measurements.As conclusion it is better to highlight this important topic again.

Reviewer 2 Report

Comments and Suggestions for Authors

Review

“Contributions Regarding the Study of Pulsatility and Resistivity Indices of Uterine Arteries in Term

Pregnancies—A Prospective Study in Bucharest, Romania”

Thank you for allowing me to review the manuscript entitled “Contributions Regarding the Study of

Pulsatility and Resistivity Indices of Uterine Arteries in Term Pregnancies—A Prospective Study in

Bucharest, Romania,” This study aims to analyze the possible correlation between PI and the RI of the

uterine arteries, evaluated with Doppler flowmetry and the different pathologies and complications that can

develop during full term pregnancy.

I thank the authors for their submission and ask that they please address the following comments and

suggestions.

Introduction:

The authors need to specify what the purpose of their research is. It is not clear.

Methodology

- Sample Size and Power Analysis: The sample size was not defined a priori with an appropriate power analysis. What is the post hoc power of this study?

- Why did the authors also consider women who were taking anticoagulant or antiplatelet therapy? This is a confounding factor

-Are all pregnant patients included regardless of the obstetric pathology they present or were only some types of pathology included?

-  Did the statistical analysis take into account confounding factors? Was a multivariate analysis done?I suggest reviewing the statistical analysis

Results

Section related to results is too long, it is not clear. It would be useful to also develop a summary and summary table that includes what was discussed in the analysis of the individual pathologies. This would allow the reader to have a more intuitive and immediate tool and to appreciate the work done.

Discussion

Discussion must be much improved. There are no references. Authors should compare their results with what is already in the literature.

Figures

In my opinion Figure 3, 7 are not necessary

Comments on the Quality of English Language

English Language and Presentation: The English language and overall presentation should be improved.

Reviewer 3 Report

Comments and Suggestions for Authors

The introduction is good, providing enough information. The study design is good, as well as the methodology and statistical analysis. The results are good, but the p-value is missing in Tables 1 and 4. The presentation of the results is difficult to follow. The discussion is good but short. It needs to be supported with more references. The conclusion is correct but also short. The references are good, but more references should be added, especially related to the discussion.

Round 2

Reviewer 3 Report

Comments and Suggestions for Authors

The authors have accepted the suggestions. I believe the revisions have added significant value to the paper, and it should be published in full.